# Silage of Prickly Pears (*Opuntia* spp.) Juice By-Products

**DOI:** 10.3390/ani10091716

**Published:** 2020-09-22

**Authors:** Alessandro Vastolo, Serena Calabrò, Monica Isabella Cutrignelli, Girolamo Raso, Massimo Todaro

**Affiliations:** 1Department of Veterinary Medicine and Animal Production, University of Federico II, via Federico Delpino 1, 80137 Napoli, Italy; alessandro.vastolo@unina.it (A.V.); scalabro@unina.it (S.C.); 2Azienda Sanitaria Provinciale Agrigento Viale della Vittoria, 321-92100 Agrigento, Italy; gino_raso@libero.it; 3Department of Agricultural, Food and Forest Science (SAAF), University of Palermo, Viale delle Scienze 13, 93011 Palermo, Italy; massimo.todaro@unipa.it

**Keywords:** storage, micro silos, in vitro gas production, straw

## Abstract

**Simple Summary:**

The cactus pear plant is a *Cactaceae* which originates from the Americas and is highly resistant to arid and hot environments. The plant is used as animal feedstuff in regions characterized by a lack of nutrient resources. In Italy, the fruits are known as prickly pears are usually eaten fresh by humans, and more recently have also started to be transformed into fruit juice. Various by-products (i.e., peel, pulp, and seeds) derived from the extraction of fruit juice are used for livestock feed due to their high amounts of fermentable structural carbohydrates. However, some of these by-products are difficult to conserve due to their high fermentability. The aim of the present study was to produce micro-silages of prickly pear by-products with different level of wheat straw (0, 5, and 10% as fed) and evaluate their chemical characteristics and fermentation kinetics using the in vitro gas production technique. Both chemical parameters (higher crude protein and lower neutral detergent fibre) and in vitro fermentation data demonstrated that the silage obtained with 5% of straw as the best-preserved (lower pH and ammonia nitrogen concentration).

**Abstract:**

Cactus pear cladodes are used as forage in the most arid regions. In Italy, the human consumption of prickly pear fruits and juice is gradually increasing for their numerous health benefits. In manufacturing plants that produce prickly pear juice, several by-products (prickly pear by-products PPB) are obtained. Despite their interesting nutritional characteristics, PPB are not very usable because of their poor shelf-life which is related to their high moisture and sugar content. The aim of this study was to verify the efficacy of ensilage to preserve PPB and to compare different inclusion levels (0, 5, and 10% as fed) of wheat straw. For each treatment, four under vacuum micro-silos were prepared and, after 35 days of storage, the state of preservation was evaluated. Subsequently, the aliquots were analyzed for chemical composition and incubated with bovine rumen fluid to evaluate the fermentation kinetics. The PPB 5% of straw showed significant lower pH and ammonia nitrogen concentration, indicating a better preservation process. Moreover, PPB 5% of straw showed better nutritional parameters (higher crude protein and lower Neutral Detergent Fibre) and fermentation characteristics (higher degradability and VFA volatile fatty acids production) when compared with the other PPB silages. Ensilage with straw represents a suitable storage technique to preserve the nutritional characteristics of PPB.

## 1. Introduction

The cactus pear plant (*Opuntia* spp.), which belongs to the *Cactaceae* family, originates from the Americas and it is cultivated in many countries around the world [1]. Cactus pear is considered an excellent and resistant plant, well adapted to arid and hot environments. Furthermore, the adaptation over millennia to these adverse environments determined the efficiency of the cactus pear in converting water to dry matter [2]. Thanks to its morphology, this species is considered a highly valuable source of nutrients in many of the world’s environments and is considered as valuable feedstuff in regions where other plants are not able to survive due to extreme environmental conditions [3]. Recently, many studies have been carried out to evaluate the effect of *Opuntia* cladodes as forage on livestock performance and rumen physiology [4,5,6,7].

Italy is the third-largest producer in the world of the *Opuntia* fruits, commonly named prickly pears, after Mexico and the United States. In Italy, the prickly pear fruits have always been consumed but in recent years, thanks to the highlighting of its numerous health benefits, the human consumption of the fruit and its juice has gradually increased. Prickly pear fruit is considered a functional food with recognized antioxidant properties [8], mainly due to the ascorbic acid, polyphenols, and flavonoids compounds which it contains [9,10]. Sicily is the Italian region which produces most prickly pear fruits (85% of national production), with a total production of 146,987 tons/year [11]. In this region, there are several manufacturing plants where the fruits are processed in order to extract the juice, and as a consequence, various by-products (i.e., peel, pulp, and seeds) are obtained. Some authors have evaluated the nutritional characteristics of this by-product, underlining the moderately high ether extract value, amounts of crude protein, fiber, and sugar [12]. However, the poor shelf-life of these products due to their high level of moisture and fermentable carbohydrates has also been evidenced.

In view of the environmental changes being experienced, the use of some of these by-products could be a way to satisfy animal needs, while, at the same time, making their production more sustainable and reducing waste production [13,14,15]. The by-products of prickly pears (PPB) are mainly managed in a fresh state, and outdoor storage is only possible for a few days due to bacterial fermentations [12].

Ensiling could be a valid way of conserving these by-products for a long period due to the anaerobic fermentation process which it involves. This storage method has also been suggested as a way of preserving *Opuntia* cladodes and fruit [16,17]. However, since PPB show high levels of moisture, it is advisable to ensile them with dry forages or mature crop residues, such as wheat straw, in order to partially absorb the water and to balance water-soluble carbohydrates and nitrogen fractions [18]. There is no research information on PPB silage referring to nutritional aspects and in vitro fermentation characteristics.

In order to study feedstuff preservation in laboratory-scale silos, various systems have been developed [19,20,21]. The use of vacuum-packed polyethylene bags suggested by Johnson et al. [21] is particularly flexible, repeatable, and cheap. Moreover, this method allows different aliquots to be obtained and the comparison of different treatments, such as the inclusion of additives or other feedstuffs.

The evaluation of fermentation characteristics with in vitro cumulative gas production technique proposed by Theodorou [22], in addition to the chemical evaluation, allows the nutritional characteristics of a novel feedstuff to be defined. More particularly, the evaluation of fermentation kinetics and the determination of the end-products (i.e., ammonia, volatile fatty acids) are useful in order to predict the rumen fermentation pathway.

The aim of this study was to verify the efficacy of ensilage as a conservation method for PPB comparing different inclusion levels (0, 5, and 10% as fed) of wheat straw. For this purpose, the nutritional characteristics and the in vitro fermentation characteristics and kinetics parameters of the silage have been studied.

## 2. Materials and Methods

### 2.1. Micro Silos Preparation

The ensiling process at laboratory scale was employed in this study, in accordance with Johnson et al. [21]. A commercial chamber vacuum-packing machine (Lavezzini device; Fiorenzuola d’Arda, Piacenza, Italy) was used to remove air from the bag equipped with an automatic heat-sealing mechanism that seals the bag after air extraction.

At the end of August 2019, 45 kg of PPB was taken directly from a prickly pear juice extraction factory in the province of Palermo (Sicily) Italy and transferred in the experimental laboratories of the Department of Agricultural, Food and Forest Science, University of Palermo, Italy, where three samples were prepared: adding to PPB chopped (2 cm) wheat straw in ratio of 0, 5, and 10% on a fresh weight basis. For each treatment, four polyethylene bags were filled with 500 g of PPB and/or straw. The bags (400 × 500 mm) were made of polyamide bioriented (OPA) and polypropylene (PP) (15 µm OPA/75 µm PP) and were characterized by an oxygen permeability of 30 cm^3^; the air vacuum pump drawed air at 10 m^3^ h^−1^ at 25 °C (Alpak srl, Taurisano, Italy). The vacuum bag silos were stored in a conditioned room (at 18 °C) for 35 days. Subsequently each bag was opened, and chemical, physical, and in vitro gas production analysis were performed.

### 2.2. Evaluation of Chemical Composition and Silage Quality

The 12 silage samples together with straw were analyzed according to the procedures of the Association of Official Agricultural Chemists (AOAC) [23] to determine dry matter (DM, 934.01), ether extract (EE, 920.39), crude protein (CP, 2001.11) and ash (942.05). The fiber fractions Neutral Detergent Fiber NDF on organic matter basis (NDFom, 2002.04), Acid Detergent Fiber on organic matter basis (ADFom, 973.18) and Acid Detergent Lignin (973.18) were determined in accordance with AOAC [23] and Van Soest et al. [24] and expressed exclusive of residual ash. To study the micro-silos quality, ammonia nitrogen (N-NH_3_) was determined in the silage juice following the procedure of the official method of analysis [25] and pH was measured directly using a pH-meter (HI 9025 142) equipped with a spear electrode FC 200 (Hanna Instruments Inc., Woonsocket, RI, USA).

### 2.3. In Vitro Gas Production

The fermentation characteristics and kinetics were studied using the in vitro gas production technique as proposed by Theodorou [22]. All samples were weighed (1.0005 g ± 0.0003) in three replications by two gas-runs and were incubated at 39 °C under anaerobic conditions in 120 mL serum bottles, to which 74 mL of anaerobic buffer were added with rumen fluid [26]. The rumen fluid was collected in a pre-warmed thermos at a slaughterhouse authorized according to EU legislation [27] from six fasting bovine (*Bos taurus*) young bulls fed a standard diet (% DM: NDF 45.5 and CP 12.0). The collected rumen content was rapidly transported to the laboratory of the Department of Veterinary Medicine and Animal Production (University of Napoli, Italy), where it was pooled, flushed with CO_2_, filtered through a cheesecloth, and added to each bottle in ratio of 10 mL. For each gas-run three bottles without substrate were incubated as blanks to correct for the disappearance of organic matter (OM) and the production of gas and end-products. Gas production of the fermenting cultures was recorded 21 times (from 2 to 24 h intervals) during the period of incubation using a manual pressure transducer (Cole and Palmer Instrument Co, Vernon Hills, IL, USA). The fermentation was stopped at 120 h by cooling at 4 °C and the fermentation liquor was analyzed for pH using a pH-meter (ThermoOrion 720 A+, Fort Collins, CO, USA) and sampled for end-product analysis. The extent of sample disappearance, expressed as organic matter degradability (OMD, %), was determined by weight difference of the incubated OM and the undegraded filtered (sintered glass crucibles; Schott Duran, Mainz, Germany, porosity # 2) residue burned at 550 °C for 3 h. The cumulative volume of gas produced after 120 h of incubation was related to incubated OM (OMCV, mL/g).

Regarding the determination of volatile fatty acids (VFA), the fermentation liquor was centrifuged at 12,000 g for 10 min at 4 °C (Universal 32R centrifuge, Hettich FurnTech Division DIY, Melle-Neuenkirchen, Germany) and 1 mL of supernatant was then mixed with 1 mL of oxalic acid (0.06 mol). The VFAs were measured by gas chromatography (ThermoQuest 8000top Italia SpA, Rodano, Milan, Italy; fused silica capillary column 30 m, 0.25 mm ID, 0.25 μm film thickness), using an external standard solution composed of acetic, propionic, butyric, iso-butyric, valeric and isovaleric acids [26].

All procedures involving animals were approved by the Ethical Animal Care and Use Committee of the University of Napoli Federico II (Prot. 2019/0013729 of 08/02/2019).

### 2.4. Statistical Analysis

For each bottle, the gas production profiles were processed with a sigmoid model described by Groot et al. [28]:(1)$$G=\frac{A}{1+{\left(\frac{B}{t}\right)}^{c}}$$
where *G* is the total gas produced (mL/g of OM) at time t (h), *A* is the asymptotic gas production (mL/g of OM), *B* (h) is the time at which one-half of the asymptote is reached, and *C* is the switching characteristic of the curve. Maximum fermentation rate (*R_max_*, mL/h) and the time at which it occurred (*T_max_*, h) were also calculated according to the following formulas [29]:(2)$$R_{max}=\frac{A*B^{C}\times B\times T_{max}{}^{\left(B-1\right)}}{\left[1+{\left(C^{B}\times T_{max}{}^{-B}\right)}^{2}\right]}$$
(3)$$T_{max}=C\times {\left[\frac{B-1}{B+1}\right]}^{1/B}.$$

The GLM and CANDISC procedures of the SAS software package version 9.2 [30] were used for the statistical analysis. Chemical characteristics, silage quality, and in vitro fermentation data were analyzed by GLM procedure for repeated measures, with the effect of substrate (PPB with 0, 5, and 10% of straw added; only straw) as the principal factor. When a significant effect (*p* < 0.05) was detected, Tukey’s test was used for means comparisons. A multivariate statistical approach was performed by a canonical discriminant analysis according to the CANDISC procedure, in order to ascertain the ability of chemical characteristics, silage quality and in vitro fermentation parameters to discriminate between the different samples. The general objective of Canonical Discriminant Analysis (CDA) is to distinguish among different populations using a particular set of variables [31]. Unlike cluster analysis, in CDA, the group to which each individual belongs is known. In this study, CDA was applied to discriminate feed substrate using chemical, silage quality and in vitro gas production parameters. Given the classification criterion (the substrate), CDA derives a new set of variables, the canonical functions (CAN), which are linear combinations of the original markers. The coefficients of the linear combinations are the canonical coefficients (CC), which indicate the partial contribution of each original variable. In this study, two canonical functions (CAN1 and CAN2) were derived. The statistical significance in group separation can be expressed by means of the Mahalanobis distance and the corresponding Hotelling’s T-square test. Groups are declared significantly separated if the Hotelling’s test shows a *p*-value < 0.05. This test can be developed only if the pooled (co)variance matrix of data is not singular. However, visual inspection of the CAN1 vs. CAN2 scatter plot and the values of distances among groups can be useful in assessing if groups are separated. CDA and the related tests were developed using the CANDISC procedure implemented in SAS software [30].

## 3. Results

### 3.1. Micro Silos Quality and Composition

In Table 1 the parameters to evaluate the silage quality (pH, DM, N-NH_3_) and the chemical composition of different substrates, including wheat straw are reported. The silage dry matter content varied from 26.46 to 28.42%, in PPB 5% and PPB 10%, respectively. The values of pH registered at the end of storage period varied from 3.85 to 3.99 in silages at 5 and 0% of straw inclusion, respectively. The ammonia nitrogen of residual fluid was below 16% in all the theses. The effect of substrate resulted significant (*p* < 0.001) for all chemical parameters considered. In particular, the straw showed mean values significantly (*p* < 0.01) different compared to all PPB silage, with the exception of ether extract. In particular, crude protein and ash contents were lower in straw, while among the structural carbohydrates, straw cellulose, and hemicellulose values resulted higher than silage samples, and an opposite result was shown for lignin. Observing PPB silage parameters, few significant differences appeared: only PPB silage with 5% of straw inclusion showed the lowest (*p* < 0.01) value for NDFom while the lignin content significantly (*p* < 0.01) decreased as the percentage of added straw increased.

### 3.2. In Vitro Fermentation Characteristics

In Table 2, the in vitro fermentation characteristics are reported. The OMD and gas produced after 120 h of incubation (OMCV and A) were significantly (*p <* 0.01) lower for all silage samples compared to straw; for all these parameters, the lowest value (*p* < 0.01) was observed in PPB silage without straw added. Regarding the in vitro fermentation kinetics, PPB silage without straw presented the highest values (*p <* 0.01) for R_max_ and the lowest values for B and T_max_, whereas these last parameters resulted higher for straw (*p <* 0.01). The differences between substrates in the fermentation process are clearer in Figure 1 and Figure 2, where gas production rate and in vitro fermentation rate over time are shown. The curve of PPB 0% of straw reached the asymptote immediately (T_max_: 1.92 and R_max_ 7.02; *p*
*<* 0.01) and the fermentation process rapidly decreased after only 24 h of incubation. On the contrary, the curve related to straw fermentation was completely different; it reached half of the asymptote later (B: 19.60 h; *p <* 0.01) and the fermentation process continued for the 120 h of incubation. The gas production kinetics obtained incubating the PPB silages with 5 and 10% of straw showed a similar profile and, after 12 h of incubation, the process was in between the other two substrates.

Table 3 reports pH and end-products of fermentation registered after 120 h of incubation. For all PPB substrates, pH values were statistically higher (*p* < 0.01) than straw, with values higher than 6.60. For all incubated substrates, the VFA production was mainly due to the sum of acetate and propionate. The incubation of straw, compared to all the tested silages, produced the highest (*p <* 0.01) concentration of all VFA with the exception of valerate, which was higher for PPB 5%. Comparing the silages, PPB with 5% of straw showed significantly higher production of all VFA (*p <* 0.01) except for butyrate. The PPB silage without straw showed the highest proportion of BCFA.

The multivariate statistical analysis results are shown in Table 4 and Table 5 and Figure 3. Multivariate analysis confirms results of univariate analysis described. Moreover, these results allow a clear discrimination of different substrates in function of their chemical characteristics and in vitro fermentation parameters (Figure 3). Mahalanobis distances were statistically different between all centroids, separating the straw from the silage samples in a more marked way (Table 4).

In Table 5, the correlations between canonicals and original variables are reported. About 94% of the total variance was explained by the first canonical variable (CAN 1) which was positively correlated with ash, CP, lignin and valerate, while it was negatively correlated with cellulose, hemicellulose, OMD, T_max_ and all VFA with the exception of valerate. The second canonical variable (CAN 2) explain 5% of total variance and it was positively correlated with R_max_ and BCFA and negatively correlated with OMCV. Only ether extract and pH after incubation parameters presented very low correlation coefficients with canonicals.

## 4. Discussion

Regarding chemical composition, data related to straw are in line with data present in literature [32]. For prickly pears silage no data exist, but, in any case, our data were in line with those regarding fresh prickly pears by-products observed in a previous study [12]. The addition of straw guaranteed the preservation of higher soluble carbohydrates and CP in PPB silages, indicating a potential reduction in losses due to leachate. As regards the parameters considered to test the silage quality (DM and N-NH_3_), the results showed good silage processing [33] for PPB, especially when 5% of straw is added (significantly lower pH and ammonia nitrogen concentration).

Only a few data are present in literature on in vitro gas production for *Opuntia* spp. Batista [34] compared three cactus varieties (Gigante, IPA-20 and Miùda) using the in situ degradability and in vitro gas production techniques. They observed differences among species for chemical composition (CP and NDF; *p <* 0.05) and potential gas production after 48 h of incubation which was significantly higher for Gigante than for the Miuda or IPA-20 varieties, despite a similar lag-phase in gas production. They reported R_max_ and a potential gas production values close to that registered in this study. In our study, comparing the three PPB silages for the in vitro fermentation characteristics, the inclusion of 5% of straw seems to guarantee suitable chemical parameters (higher CP and ether extract content; lower NDF and ADF values) which contribute to the higher digestibility and VFA production.

The results of the multivariate statistical analysis confirmed that of univariate analysis. Indeed, the canonical discriminant analysis ascertained the ability to discriminate the substrates (*p <* 0.001). Particularly interesting were the correlation coefficients shown between canonicals and original variables. The first canonical variable, that explained most of the variability, seems to be linked to fermentation of the substrate into the rumen. The straw alone is placed on the left of the plot (Figure 3) contrary to the ensiled substrates which have higher values on CAN 1 (on the right of the plot). The latter seem to be less fermentable than straw, as they are associated with higher contents of ash (r = +0.890), lignin (r = +0.908) and proteins (r = +0.976) and consequently determine a lower quantity of VFA, correlated almost all negatively with CAN 1. The positive correlation between CP and CAN 1, together with lignin and ash, could be explained by a possible interference of some bioactive molecules, such as condensed tannins in chemical measurement of lignin using the conventional gravimetric method [35,36]. Indeed, Marles [36] observed that in presence of condensed tannins the ADF-ADL technique overestimate the lignin content in comparison to alternative techniques such as thioglycolic acid lignin assay. On the other hand, the possible presence of condensed tannins in PPB [9,15] could reduce the rumen fermentability. As regards the gas production parameters, OMD and T_max_ resulted negatively correlated with the first canonical, and this fact showed that straw fermentation is slower than silages. The second canonical variable is able to discriminate between the silage with straw (5 and 10%) from that without straw, which ranks higher in the plot among the canonical discriminants (Figure 3) and is associated with the highest values of CAN 2. The gas production parameters—R_max_ (r = +0.594) and OMCV (r = −0.712)—are more capable of discriminating PPB silage. The presence of straw in the silage, especially in the amount of 5%, determines a more intense fermentative process, in terms of higher values of OMD, volume of gas, and VFA production (mainly acetate and propionate) but slowed down the fermentation kinetics (lower R_max_ and higher B and T_max_ values for PPB 5% compared to PPB 0% and 10%). These results could be due to a ‘dilution effect’ of the components (ether extract, lignin, ash) that can reduce the extent of the in vitro fermentations. On the other hand, a major presence of fermentable carbohydrates in silage without straw may have favored a quick start of the fermentation process. Moreover, the absence of straw in PPB silage determines a higher BCFA (r = +0.563) production probably due to a major degradation of protein, and consequently a higher production of BCFA deriving from some amino acids [37] during the ensilage process.

## 5. Conclusions

From these preliminary results it is possible to demonstrate that ensilage is a suitable storage technique to preserve the nutritional characteristics of PPB. In particular, the addition of wheat straw to PPB seems useful as a way of reducing nutrient losses during ensiling. Both chemical parameters and in vitro fermentation data indicated the silage obtained with 5% of straw as the best preserved. This observation could also denote an economical advantage for the lower level of straw inclusion. In order to evaluate the palatability of PPB silages and the effects of their administration to dairy cows or dairy ewes on milk yield and quality, further studies are necessary.

## Figures and Tables

**Figure 1 animals-10-01716-f001:**
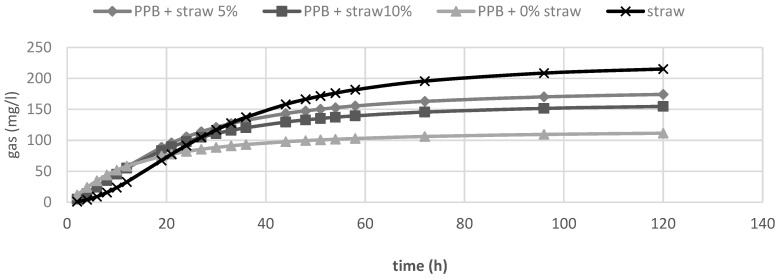
In vitro gas production over time of prickly pear by-products (PPB) silages and wheat straw.

**Figure 2 animals-10-01716-f002:**
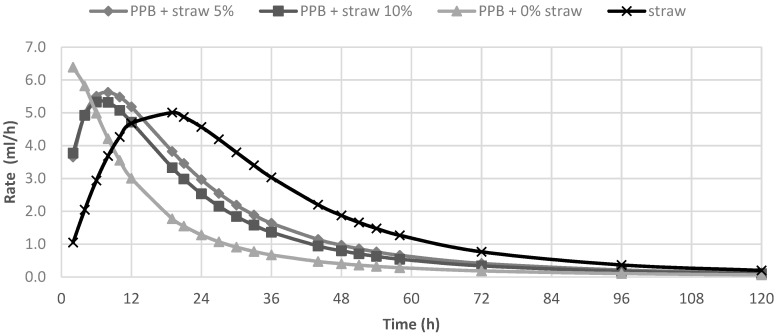
In vitro fermentation rate of prickly pear by-products (PPB)silages and wheat straw.

**Figure 3 animals-10-01716-f003:**
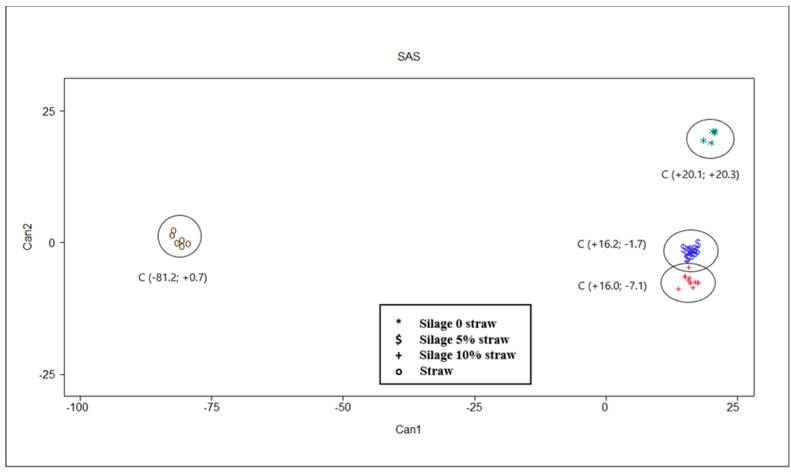
Plot from canonical discriminant analysis in which different feeds are distributed in function of canonical variables 1 and 2 based on chemical, silage quality and in vitro gas production parameters. Centroid coordinates in parentheses are reported.

**Table 1 animals-10-01716-t001:** Micro silos evaluation and chemical composition of by-products of prickly pears (PPB) silages and wheat straw.

Parameters		PPB Silage: Straw Percentages			Straw	Substrate
		0%	5%	10%		*p* Value
DM	%	27.68 ± 0.42 ^B^	26.46 ± 0.27 ^B^	28.42 ± 0.27 ^B^	91.95 ± 0.38 ^A^	<0.001
pH		3.99 ± 0.01 ^A^	3.85 ± 0.01 ^C^	3.96 ± 0.01 ^B^	Nd	<0.001
N-NH_3_	% TN	14.10 ± 0.18 ^B^	13.29 ± 0.12 ^C^	15.24 ± 0.12 ^A^	Nd	<0.001
CP	% DM	6.91 ± 0.14 ^Aa^	6.68 ± 0.09 ^Aab^	6.40 ± 0.09 ^Bb^	2.85 ± 0.13 ^C^	<0.001
Ether extract	“	6.13 ± 0.39 ^AB^	6.97 ± 0.25 ^A^	5.30 ± 0.25 ^B^	7.01 ± 0.35 ^A^	<0.001
NDFom	“	61.12 ± 0.38 ^B^	59.50 ± 0.24 ^C^	60.69 ± 0.24 ^B^	82.13 ± 0.34 ^A^	<0.001
ADFom	“	48.39 ± 0.51 ^B^	48.57 ± 0.33 ^B^	49.34 ± 0.33 ^B^	54.66 ± 0.47 ^A^	<0.001
Hemicellulose	“	12.73 ± 0.44 ^B^	10.94 ± 0.29 ^C^	11.34 ± 0.29 ^C^	27.48 ± 0.40 ^A^	<0.001
ADL	“	14.68 ± 0.21 ^A^	13.86 ± 0.14 ^B^	12.98 ± 0.14 ^C^	9.46 ± 0.19 ^D^	<0.001
Cellulose	“	33.71 ± 0.40 ^Cd^	34.70 ± 0.26 ^Cc^	36.36 ± 0.26 ^B^	45.19 ± 0.37 ^A^	<0.001
Ash	“	10.26 ± 0.22 ^A^	10.20 ± 0.14 ^A^	10.19 ± 0.14 ^A^	7.83 ± 0.20 ^B^	<0.001

PPB: Prickly pear by-products. DM: Dry matter. NH_3_-N (% TN): Ammonia nitrogen expressed as Total Nitrogen.CP: Crude Protein; NDFom: Neutral Detergent Fibre on organic matter basis; ADFom: Acid detergent fibre on organic matter basis; ADL: Acid Detergent Lignin; Nd: Not determined. Along the row different capital superscript letters indicate difference for *p <* 0.01; different lowercase superscript letters indicate difference for *p <* 0.05.

**Table 2 animals-10-01716-t002:** Cumulative gas production, organic matter degradability and fermentation kinetics parameters of PPB silages and wheat straw.

Parameter		PPB Silage: Straw Percentages			Straw	Substrate
		0%	5%	10%		*p* Value
OMD	%	45.09 ± 1.00 ^D^	56.10 ± 0.65 ^B^	50.36 ± 0.65 ^C^	60.36 ± 0.91 ^A^	<0.001
OMCV	mL/g	132.68 ± 5.28 ^C^	206.53 ± 3.41 ^B^	211.93 ± 3.41 ^B^	252.33 ± 4.82 ^A^	<0.001
A	mL/g	120.92 ± 4.09 ^D^	184.52 ± 2.64 ^B^	163.57 ± 2.64 ^C^	227.78 ± 3.73 ^A^	<0.001
B	h	12.35 ± 0.47 ^C^	18.59 ± 0.31 ^B^	29.24 ± 0.43 ^B^	19.60 ± 0.31 ^A^	<0.001
C		1.25 ± 0.06 ^C^	1.61 ± 0.04 ^B^	1.53 ± 0.04 ^B^	2.00 ± 0.06 ^A^	<0.001
R_max_	mL/h	7.32 ± 0.30 ^A^	5.80 ± 0.23 ^B^	5.38 ± 0.23 ^B^	5.07 ± 0.33 ^B^	<0.001
T_max_	h	1.92 ± 0.55 ^D^	7.91 ± 0.35 ^B^	6.32 ± 0.35 ^C^	16.84 ± 0.50 ^A^	<0.001

PPB: Prickly pear by-products. OMD: Organic matter disappearance; OMVC: Cumulative volume of gas related to incubated organic matter; A: asymptotic gas production, B: is the time at which one-half of the asymptote is reached; C: switching characteristic of the curve; R_max_: maximum fermentation rate; T_max_: time at which R_max_ occurs. Along the row different capital superscript letters indicate difference for *p <* 0.01; different lowercase superscript letters indicate difference for *p <* 0.05.

**Table 3 animals-10-01716-t003:** PH and in vitro fermentation end-products after 120 h of incubation of PPB silages and wheat straw.

Parameter		PPB Silage: Straw Percentages			Straw	Substrate *p* Value
		0%	5%	10%		
pH		6.86 ± 0.07 ^A^	6.75 ± 0.04 ^A^	6.87 ± 0.04 ^A^	6.63 ± 0.06 ^B^	<0.001
Acetate	mmol/g OM	34.75 ± 1.60 ^D^	46.17 ± 1.04 ^B^	42.51 ± 1.00 ^C^	59.64 ± 1.47 ^A^	<0.001
Propionate	“	11.24 ± 0.51 ^D^	17.52 ± 0.33 ^B^	15.24 ± 0.33 ^C^	21.90 ± 0.47 ^A^	<0.001
Iso-butyrate	“	0.35 ± 0.01 ^C^	0.39 ± 0.01 ^B^	0.32 ± 0.01 ^C^	0.49 ± 0.01 ^A^	<0.001
Butyrate	“	3.98 ± 0.28 ^Bc^	4.60 ± 0.18 ^Bb^	3.95 ± 0.18 ^Bc^	6.50 ± 0.25 ^Aa^	<0.001
Iso-valerate	“	0.49 ± 0.03 ^B^	0.67 ± 0.02 ^A^	0.48 ± 0.02 ^B^	0.75 ± 0.03 ^A^	<0.001
Valerate	“	0.76 ± 0.05 ^B^	1.03 ± 0.03 ^A^	0.75 ± 0.03 ^B^	0.54 ± 0.05 ^C^	<0.001
VFA	“	52.78 ± 2.19 ^D^	70.38 ± 1.41 ^B^	63.75 ± 1.41 ^C^	89.80 ± 2.00 ^A^	<0.001
BCFA	% VFA	1.62 ± 0.06 ^Aa^	1.52 ± 0.04A ^Ba^	1.28 ± 0.04 ^Cc^	1.39 ± 0.05B ^Cb^	<0.001

PPB: Prickly pear by-products. VFA: Volatile fatty acids; BCFA: Branched-chain fatty acids (iso-valerate + iso-butyrate/total VFA × 100); OM: Organic Matter. Along the row different capital superscript letters indicate difference for *p* < 0.01; different lowercase superscript letters indicate difference for *p* < 0.05.

**Table 4 animals-10-01716-t004:** Canonical discriminant analysis: Mahalanobis quadratic distances.

Substrate	PPB 0% Straw	PPB 5% Straw	PPB 10% Straw	Straw
PPB 0% straw	0	49 (*p* < 0.001)	69 (*p <* 0.001)	729 (*p* < 0.001)
PPB 5% straw		0	15 (*p <* 0.001)	955 (*p* < 0.001)
PPB 10% straw			0	956 (*p <* 0.001)
Straw				0

PPB: Prickly pear by-products.

**Table 5 animals-10-01716-t005:** Total canonical structure: Correlations between canonicals and original variables.

Variable	1st Canonical Variable	2nd Canonical Variable
Ash	**0.890**	−0.007
CP	**0.976**	0.069
Ether extract	−0.293	0.168
Cellulose	**−0.954**	−0.164
Hemicellulose	**−0.981**	0.115
Lignin	**0.908**	0.321
pH	0.419	0.011
OMD	**−0.623**	−0.375
OMCV	−0.622	**−0.712**
T_max_	**−0.888**	−0.310
R_max_	0.321	**0.594**
Acetate	**−0.805**	−0.319
Propionate	**−0.746**	−0.418
Butyrate	**−0.795**	0.004
Iso-butyrate	**−0.778**	0.081
Valerate	**0.585**	−0.089
Iso-valerate	**−0.556**	−0.060
BCFA	0.123	**0.563**
*Explained variance (%)*	*93.8*	*5.3*

PPB: Prickly pear by-products; CP: Crude Protein; OMD: Organic matter disappearance; OMVC: Cumulative volume of gas related to incubated organic matter; R_max_: Maximum fermentation rate; T_max_: Time at which R_max_ occurs, BCFA: Branched-chain fatty acids (iso-valerate + iso-butyrate/total VFA × 100). In bold the heavier correlation coefficients are reported.

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
