# Peer review of "Silage of Prickly Pears (*Opuntia* spp.) Juice By-Products"

_animals, 2020, doi:10.3390/ani10091716_

Round 1

Reviewer 1 Report

Language revision is needed. The English is often very “Italian”: for instance, the use of “the” or not, sentence constructions (example: line 29-30), incorrect plural or singular of verbs and nouns, using plural before a noun (e.g. “prickly pears fruit instead” of “prickly pear fruit”), etcetera. For readability purposes this manuscript therefore requires linguistic revision by a professional copywriter or native English-speaking academic. I have made some suggestions but stopped correcting at a certain point.

15: Please rephrase as “The cactus pear plant is a Cactaceae originating from the Americas.”

15: “highly”

15-16: “cactus” and “regions” without capital

33: “thesis” : do you mean “treatment”?

36: write “N-NH3 values” as ‘ammonia nitrogen concentration”

37: “better” is an interpretation; this depends on the purpose: more ether extract is not always beneficial for rumen function whereas structural carbohydrates are needed for safe fermentation

46-48: I have no idea what this sentence means.

56: “prickly pear fruit”: as an example here, no plural is written when followed by another noun. This has to be checked throughout the manuscript.

69: rewite “The silage making” as “Ensiling”

73-74: I miss the state-of-the-art on ensiling of Opuntia: there is quite some published information on this topic, so the authors need to clarify better what exactly is the novelty. There is for instance a study on the nutritive value of ensiled Opuntia with teff straw.  

90: please explain how the straw was homogenously added: was this ground straw or chopped? Particle size? And straw from which plant?

117: “flasks”

169: (comment for ALL tables and figures): the titles should be stand alone so that a reader can understand without reading the manuscript itself.

Table 1:

- it seems odd to include pure straw in the statistical comparison with the 0, 5, 10% additions. I do not see the value of that treatment in optimizing PPB ensiling.

Figure 2: was there no time zero measurement? It seems that the peak of the 0% straw treatment was missed…

- the combination of lower and higher case superscripts is confusing and reduces readability; only use normal superscripts for P<0.05). Using a t-test as post-hoc test is also unusual. Instead use something like Scheffé or Tukey.

- P-values cannot be exactly 0.001 all the time: I think the authors need to write “<0.001”.

Table 2: please discuss the potential explanation for the non-linearity of “A” and “Tmax”

Table 4, Table 5 and Figure 3 are somehow redundant: they are invalid because of the intrinsic cohesion of parameters in only four treatments and to some extent they are also duplicating each other. They do not add further insights. This type of analysis is meant to be used for large datasets with numerous parameters, so this approach is a bit an overstretch here, and is best removed.

Author Response

Answer referee 1

Dear Reviewer,

Thank you for your review. Yours suggestion helped us to improve the scientific value of our manuscripts. All the corrections have been marked in red.

Comments and Suggestions for Authors

Language revision is needed. The English is often very “Italian”: for instance, the use of “the” or not, sentence constructions (example: line 29-30), incorrect plural or singular of verbs and nouns, using plural before a noun (e.g. “prickly pears fruit instead” of “prickly pear fruit”), etcetera. For readability purposes this manuscript therefore requires linguistic revision by a professional copywriter or native English-speaking academic. I have made some suggestions but stopped correcting at a certain point.

RE Authors: The paper was revised by an English native lecturer

15: Please rephrase as “The cactus pear plant is a Cactaceae originating from the Americas.”

RE Authors: Done

15: “highly”

RE Authors: Done

15-16: “cactus” and “regions” without capital

RE Authors: Done

33: “thesis”: do you mean “treatment”?

RE Authors: Done

36: write “N-NH3 values” as ‘ammonia nitrogen concentration”

RE Authors: Done

37: “better” is an interpretation; this depends on the purpose: more ether extract is not always beneficial for rumen function whereas structural carbohydrates are needed for safe fermentation

RE Authors: Done

46-48: I have no idea what this sentence means.

RE Authors: the sentences were re-written according to the other referee

56: “prickly pear fruit”: as an example here, no plural is written when followed by another noun. This has to be checked throughout the manuscript.

RE Authors: Done

69: rewite “The silage making” as “Ensiling”

RE Authors: Done

73-74: I miss the state-of-the-art on ensiling of Opuntia: there is quite some published information on this topic, so the authors need to clarify better what exactly is the novelty. There is for instance a study on the nutritive value of ensiled Opuntia with teff straw.

RE Authors: In literature ensiling was proposed to preserve cladodes or fruit. In this study we evaluate the use of ensiling for prickly pear fruit by-products. The sentence was changed to clarify this point

90: please explain how the straw was homogenously added: was this ground straw or chopped? Particle size? And straw from which plant?

RE Authors: the sentence was re-written

117: “flasks”

RE Authors: the term was substitute by bottles

169: (comment for ALL tables and figures): the titles should be stand alone so that a reader can understand without reading the manuscript itself.

RE Authors:  the titles were changed

Table 1:- it seems odd to include pure straw in the statistical comparison with the 0, 5, 10% additions. I do not see the value of that treatment in optimizing PPB ensiling.

RE Authors: Straw has been included in the analytical and gas production evaluation process because it is the added feed for making the substrate (PPB 5% and PPB 10%) and allows better understanding of the results of the different levels of straw integration

Figure 2: was there no time zero measurement? It seems that the peak of the 0% straw treatment was missed-

RE Authors: The first gas measurement is after 2 h of incubation. There are no missing data, only the fermentation for PPB 0% straw start very rapidly.

the combination of lower and higher case superscripts is confusing and reduces readability; only use normal superscripts for P<0.05). Using a t-test as post-hoc test is also unusual. Instead use something like Scheffé or Tukey.

RE Authors: In the international literature there are a lot of papers that report lower and higher case superscripts to highlight different levels of significance at P<0.01 and P<0.05. As regards the test used we are in agreement with Referee and we used the Tukey’s test. We have corrected the paper

- P-values cannot be exactly 0.001 all the time: I think the authors need to write “<0.001”.

RE Authors: We are in accordance with referee and the tables 1, 2 and 3 were corrected

Table 2: please discuss the potential explanation for the non-linearity of “A” and “Tmax”

RE Authors: Sorry, not clear referee comment. We don’t think needs to exist a linearity between A and Tmax. Straw is a substrate rich in structural carbohydrates that produce a lot of gas (high A value), but they are not easily fermentable, so it needs more time (high Tmax). We don’t think is necessary report this concept in the paper.

Table 4, Table 5 and Figure 3 are somehow redundant: they are invalid because of the intrinsic cohesion of parameters in only four treatments and to some extent they are also duplicating each other. They do not add further insights. This type of analysis is meant to be used for large datasets with numerous parameters, so this approach is a bit an overstretch here, and is best removed.

RE Authors: we do not agree with the referee because even if it is true that the canonical discriminant analysis is usually done by data-set with many detected parameters, the aim of this work is to evaluate the different substrates and understand if they are different considering overall the parameters of the chemical, physical composition and fermentation kinetics through the gas production technique. In addition, different published papers that used the CANDISC analysis report 29 items (Sorbolini, S., Gaspa, G., Steri, R., Dimauro, C., Cellesi, M., Stella, A., & Macciotta, N. P. P. 2016. Use of canonical discriminant analysis to study signatures of selection in cattle. Genetics Selection Evolution,48, 58.), 24 items (Dimauro, C., Cellesi, M., Steri, R., Gaspa, G., Sorbolini, S., Stella, A., & Macciotta, N. P. P. 2013. Use of the canonical discriminant analysis to select SNP markers for bovine breed assignment and traceability purposes. Animal genetics, 44, 377-382), but also 5 items (Metzger, S. A., Hernandez, L. L., Skarlupka, J. H., Suen, G., Walker, T. M., & Ruegg, P. L. 2018. Influence of sampling technique and bedding type on the milk microbiota: results of a pilot study. Journal of Dairy Science, 101, 6346-6356) or 4 items (Bava, L., Zucali, M., Sandrucci, A., Brasca, M., Vanoni, L., Zanini, L., & Tamburini, A. 2011. Effect of cleaning procedure and hygienic condition of milking equipment on bacterial count of bulk tank milk. The Journal of Dairy Research, 78, 211), in our data-set we used 18 items.

Reviewer 2 Report

The authors presented an analysis of prickly pear silage fermentation with in vitro and discriminant analysis. The manuscript has many grammatical errors that need serious revision. English revision is highly needed. The figures and tables are in good quality. The discussion is well presented; the discriminant analysis is discussed in a satisfactory manner in order to explain the correlation between by-products and the discriminant variables, however, a better explanation on the probabilities, t-Student and discriminant analysis methodology is needed.

Some few excamples

Simple Summary

Line 15: Grammar error

Line 16: Which authors? Avoid these sentences as citations are needed and are not recommended in an abstract

Line 23: What is the definition of “level of wheat straw”? What exactly the percentages refer to? Unclear sentence

Line 24: Which chemical parameters?

Line 25: What is to be considered as the “best” preserved?

Abstract

The abstract contains many grammatical and punctuation errors

Line 33: “35 d” what is “d”?

Introduction

Line 73-74: It seems that is already literature pertaining this theme. I suggest that the authors revaluate the design of the work or rather emphasize the importance of your paper with your most important findings or novel approaches and discussions

Materials and Methods

Line 151: State the null hypothesis

Line 151: “p<0.05” Is a comparison with the probability result not the significance level

Line 151-154: “A multivariate statistical approach was performed by a canonical discriminant analysis” Reference is missing. Which software and packages were used? The performance of this analysis is insufficiently described

Results

Line 162: “p<0.001” p stands for the probability of what? What is this variation (standard deviation or relative standard deviation)?

Line 157-168: Revise this section. It is highly confusing; define what those probabilities refer to

Discussion

It would be interesting to see a discussion of why 5% straw addition was better than 10%

rewrite in a more logical sequence, without going back on the subject

Author Response

Dear Reviewer,

Thank you for your review. Yours suggestion helped us to improve the scientific value of our manuscripts. All the corrections have been marked in red.

The authors presented an analysis of prickly pear silage fermentation with in vitro and discriminant analysis. The manuscript has many grammatical errors that need serious revision. English revision is highly needed. The figures and tables are in good quality. The discussion is well presented; the discriminant analysis is discussed in a satisfactory manner in order to explain the correlation between by-products and the discriminant variables, however, a better explanation on the probabilities, t-Student and discriminant analysis methodology is needed.

Some few examples

Simple Summary

Line 15: Grammar error

RE Authors: The paper was revised by an English native lecturer

Line 16: Which authors? Avoid these sentences as citations are needed and are not recommended in an abstract.

RE Authors: The sentence was changed

Line 23: What is the definition of “level of wheat straw”? What exactly the percentages refer to? Unclear sentence

RE Authors: The percentage of straw inclusion is referred to fresh weight. It was specified into the text

Line 24: Which chemical parameters?

RE Authors: The parameters are reported

Line 25: What is to be considered as the “best” preserved?

RE Authors: The lower values of pH and N-NH3 are indicative of a better ensiling process

Abstract

The abstract contains many grammatical and punctuation errors

Line 33: “35 d” what is “d”?

RE Authors: The abstract was re-written

Introduction

Line 73-74: It seems that is already literature pertaining this theme. I suggest that the authors revaluate the design of the work or rather emphasize the importance of your paper with your most important findings or novel approaches and discussions

RE Authors: The introduction was re-written according your suggestions

Materials and Methods

Line 151: State the null hypothesis

RE Authors: we prefer to change the period as follows “when a significant effect (P < 0.05) was detected, Student t test was used for means comparisons”

Line 151: “p<0.05” Is a comparison with the probability result not the significance level

RE Authors: we are in accordance with Referee and we changed the period as follows “when a significant effect (P < 0.05) was detected, Student t test was used for means comparisons”.

Line 151-154: “A multivariate statistical approach was performed by a canonical discriminant analysis” Reference is missing. Which software and packages were used? The performance of this analysis is insufficiently described

RE Authors: we add the Reference (Mardia et al., 2000); the software used is SAS CANDISC procedure, but this was reported at line 147; moreover, we have better described the analysis in the text

Results

Line 162: “p<0.001” p stands for the probability of what? What is this variation (standard deviation or relative standard deviation)?

RE Authors: we are in accordance with Referee and we changed the period as follows: “The effect of substrate resulted significant (P<0.001) for all chemical parameters considered”

Line 157-168: Revise this section. It is highly confusing; define what those probabilities refer to

RE Authors: The sentence was re-written

Discussion

It would be interesting to see a discussion of why 5% straw addition was better than 10% rewrite in a more logical sequence, without going back on the subject

RE Authors: the discussion was changed

Reviewer 3 Report

By-products could be considered a valid opportunity to satisfy animal requirement, to make the animal production more sustainable and reducing wastes.

The silage could represent a valid system to conserve the PPB for a long period due to an anaerobic fermentation process, and the lower level of wheat straw added seems to be useful to avoid nutritional losses

General comments:

- Caution with abbreviations: occasionally some of the terms can be replaced by the respective abbreviations, after the first reference. Please uniformize in all manuscript. Some of them are identified on the Specific comments.

- The references [10] and [11] seems to be exchanged, please confirm. [11] Todaro et al studied the chemical composition of prickly pears; and [10] (ISTAT) presents the production of fruits.

- The reference [12] not include "these by-products", as referred on the Introduction, they study by-products derived from citrus fruit or olive oil processing, not from prickly pears. Please clarify this sentence in accordance.

- On the Tables and text, please add space between ± and numbers.

- Please uniformize the p on the text and Table, when refers to the p values value, “p” lowercase and italic is the more correct form. And add a space between numbers and “<”.

- Please include a capital letter when refers to “Table” and “Figure”, along of the all text (Ln 157, 175 and 214, for Tables; and Ln 181, 214, 217, 259,272, for Figures). And add a point after the number on each Table and Figure caption, according to the manuscript template.

Specific comments:

Ln 49 and 58. “Region” without capital letter.

Ln 57. The authors considered the Reference 8. But, Food Chem. 2003, 83, 447–456 and Food Res. Int. 2011, 44, 2311–2318; seems to be more representative of the study on the antioxidant activity of the fruits, but it is at the discretion of the authors.

Ln 73, 287, 291. Replace “prickly pear by-products” by “PPB”.

Ln 76. Delete “prickly pear by-products”, and the brackets on “(PPB)”.

Ln 90. What are the units for the % of wheat straw added? % of dry weight or fresh weight of PPB? Please clarify.

Ln 117. Add a comma after “gas-run”

Ln 130. Replace “volatile fatty acids” by “VFA”.

Ln 161. For 10% straw, ammonia content is 15.24, so it is not lower than 15% for all. Please correct the sentence.

Ln 162. Please consider change “theses” by “studied silages”, or similar. “these” does not seem too much precise and correct.

Ln 164, 224, 243, 251, 263. Replace “crude protein” by “CP”.

Table 1. Dry matter (DM). And add “%” for PPB 0 column, and uniformize for the other tables.

Ln 176. Add “and” instead of comma “(OMCV, A)”.

Ln 180. “straw presented the highest”

Ln 181. Add comma after “Figure 1 and 2”

Ln 182. 0% of straw

Figure 1 and 2. Uniformize the capital letters on the legends

Ln 202-203. Suggestion: simplify and consider only one sentence. For example: “For all substrates, the pH was statistically significant (p < 0.01), with values higher than 6.60, after incubation.”

Ln 206. Replace “volatile fatty acids” by “VFA”. Add a comma after “valerate”. “PPB with 5% of straw”.

Ln 208. “all VFA”, except for Butyrate, please correct

Ln 225, 262, 276. Remove "s". VFA is identified as "volatile fatty acids"

Ln 230. Replace “by-products.” by “by-products;”.

Ln 236. Replace “:” by “.” after Figure 3

Ln 241. “straw” instead “sraw”.

Ln 248. Delete “data” after “in vitro gas production”

Ln 253. Replace “Rmax” by “Rmax”.

Ln 276. “OMD” instead “OM degradability”.

Ln 282. Space between “r=”.

Ln 283. “BCFA” instead “branched chain fatty acids”.

Ln 287, 291. Replace “prickly pear by-products” by “PPB”.

Author Response

REFEREE 3

Dear Reviewer,

Thank you for your review. Yours suggestion helped us to improve the scientific value of our manuscripts. All the corrections have been marked in red.

By-products could be considered a valid opportunity to satisfy animal requirement, to make the animal production more sustainable and reducing wastes.The silage could represent a valid system to conserve the PPB for a long period due to an anaerobic fermentation process, and the lower level of wheat straw added seems to be useful to avoid nutritional losse

General comments:

- Caution with abbreviations: occasionally some of the terms can be replaced by the respective abbreviations, after the first reference. Please uniformize in all manuscript. Some of them are identified on the Specific comments.

- The references [10] and [11] seems to be exchanged, please confirm. [11] Todaro et al studied the chemical composition of prickly pears; and [10] (ISTAT) presents the production of fruits.

RE Authors: The mistake was corrected thank your suggestion

- The reference [12] not include "these by-products", as referred on the Introduction, they study by-products derived from citrus fruit or olive oil processing, not from prickly pears. Please clarify this sentence in accordance.

RE Authors: It is a consequence of the previous mistake. Now it is corrected

- On the Tables and text, please add space between ± and numbers.

RE Authors: done

- Please uniformize the p on the text and Table, when refers to the p values value, “p” lowercase and italic is the more correct form. And add a space between numbers and “<”.

RE Authors: Done

- Please include a capital letter when refers to “Table” and “Figure”, along of the all text (Ln 157, 175 and 214, for Tables; and Ln 181, 214, 217, 259,272, for Figures). And add a point after the number on each Table and Figure caption, according to the manuscript template.

RE Authors: Done, thank you for the observation

Specific comments:

Ln 49 and 58. “Region” without capital letter.

RE Authors: Done

Ln 57. The authors considered the Reference 8. But, Food Chem. 2003, 83, 447–456 and Food Res. Int. 2011, 44, 2311–2318; seems to be more representative of the study on the antioxidant activity of the fruits, but it is at the discretion of the authors.

RE Authors: We added a reference that we lose during in the previous version

Ln 73, 287, 291. Replace “prickly pear by-products” by “PPB”.

RE Authors: Done

Ln 76. Delete “prickly pear by-products”, and the brackets on “(PPB)”.

RE Authors: Done

Ln 90. What are the units for the % of wheat straw added? % of dry weight or fresh weight of PPB? Please clarify.

RE Authors: The percentage of straw inclusion is referred to fresh weight. It was specified into the text

Ln 117. Add a comma after “gas-run”

RE Authors: Done

Ln 130. Replace “volatile fatty acids” by “VFA”.

RE Authors: Done

Ln 161. For 10% straw, ammonia content is 15.24, so it is not lower than 15% for all. Please correct the sentence.

RE Authors: The sentence was corrected

Ln 162. Please consider change “theses” by “studied silages”, or similar. “these” does not seem too much precise and correct.

RE Authors: The sentence was corrected

Ln 164, 224, 243, 251, 263. Replace “crude protein” by “CP”.

RE Authors: Done

Table 1. Dry matter (DM). And add “%” for PPB 0 column, and uniformize for the other tables.

RE Authors: Done

Ln 176. Add “and” instead of comma “(OMCV, A)”.

RE Authors: Done

Ln 180. “straw presented the highest”

RE Authors: Done

Ln 181. Add comma after “Figure 1 and 2”

RE Authors: Done

Ln 182. 0% of straw

RE Authors: Done

Figure 1 and 2. Uniformize the capital letters on the legends

RE Authors: Done

Ln 202-203. Suggestion: simplify and consider only one sentence. For example: “For all substrates, the pH was statistically significant (p < 0.01), with values higher than 6.60, after incubation.”

RE Authors: Done

Ln 206. Replace “volatile fatty acids” by “VFA”. Add a comma after “valerate”. “PPB with 5% of straw”.

RE Authors: Done

Ln 208. “all VFA”, except for Butyrate, please correct

RE Authors: Done

Ln 225, 262, 276. Remove "s". VFA is identified as "volatile fatty acids"

RE Authors: Done

Ln 230. Replace “by-products.” by “by-products;”.

RE Authors: Done

Ln 236. Replace “:” by “.” after Figure 3

RE Authors: Done

Ln 241. “straw” instead “sraw”.

RE Authors: Done

Ln 248. Delete “data” after “in vitro gas production”

RE Authors: Done

Ln 253. Replace “Rmax” by “Rmax”.

RE Authors: Done

Ln 276. “OMD” instead “OM degradability”.

RE Authors: Done

Ln 282. Space between “r=”.

RE Authors: Done

Ln 283. “BCFA” instead “branched chain fatty acids”.

RE Authors: Done

Ln 287, 291. Replace “prickly pear by-products” by “PPB”.

RE Authors: Done

Round 2

Reviewer 1 Report

Thank you for responding to the comments: the manuscript has greatly improved. I still doubt the (added) value of the canonical correlation analysis in this case, but do not object.

There is one small edit somewhere near the conclusion, where "preview" is written whereas likely "previous" is meant. Maybe that can be corrected in the proofing phase.

Author Response

Dear Reviewer,

Thanks for your suggestion.

In the conclusion, "preview" has changed with "previous".

Few English expression were revised.

Regards

Reviewer 2 Report

The authors expanded on the statistical methodology and improved the overall text.

Author Response

Dear Reviewer, 

thanks again for your work in improving the manuscript.

Few English expression were revised. 

Regards

Reviewer 3 Report

The reviewed version improved the manuscript.

I identified just some points that can be change:

Ln 80. "polyethylene" instead " polythene"

Results, Section 3.1. add a space after <, for the p values, both in text and table

Ln 184 and 204. "Table" instead "table"

Ln 188. delete the space after "16 %"

Ln 277. "PPB" instead "prickly pear by-products"

Author Response

Dear Reviewer, 

We have considered all the correction you suggested.

Ln 80. "polythene" has been changed in "polyethylene"

In Results, Section 3.1. a space has been added after <, for the p values, in text and table

Ln 184 and 204: "table" has been corrected with "Table"

Ln 188: the space after "16 %" has been deleted

Ln 277: "prickly pear by-products" has been corrected with "PPB" 

Few English expression were revised. 

Thanks again for your work in improving the manuscript.

Regards